# Cell-Enriched Lipotransfer (CELT) Improves Tissue Regeneration and Rejuvenation without Substantial Manipulation of the Adipose Tissue Graft

**DOI:** 10.3390/cells11193159

**Published:** 2022-10-08

**Authors:** Lukas Prantl, Andreas Eigenberger, Ruben Reinhard, Andreas Siegmund, Kerstin Heumann, Oliver Felthaus

**Affiliations:** 1Department of Plastic, Hand and Reconstructive Surgery, University Medical Center Regensburg, Franz-Josef-Strauss-Allee 11, 93053 Regensburg, Germany; 2Medical Device Lab, Regensburg Center of Biomedical Engineering (RCBE), Faculty of Mechanical Engineering, Ostbayerische Technische Hochschule Regensburg, Galgenbergstr. 30, 93053 Regensburg, Germany

**Keywords:** fat grafting, lipofilling, mesenchymal stem cells, adipose tissue-derived stem cells, regeneration, regenerative medicine, reconstructive surgery

## Abstract

The good availability and the large content of adult stem cells in adipose tissue has made it one of the most interesting tissues in regenerative medicine. Although lipofilling is one of the most frequent procedures in plastic surgery, the method still struggles with high absorption rates and volume losses of up to 70%. Therefore, many efforts have been made to optimize liposuction and to process the harvested tissue in order to increase fat graft retention. Because of their immunomodulatory properties, their cytokine secretory activity, and their differentiation potential, enrichment with adipose tissue-derived stem cells was identified as a promising tool to promote transplant survival. Here, we review the important parameters for lipofilling optimization. Finally, we present a new method for the enrichment of lipoaspirate with adipose tissue-derived stem cells and discuss the parameters that contribute to fat graft survival.

## 1. Introduction

Adipose tissue is one of the most promising sources of tissues and cells for regenerative medicine. It can be harvested with minimally invasive procedures and contains various cell types beneficial for tissue regeneration and wound healing [1,2,3]. For example, adipose tissue contains the highest number of mesenchymal stem cells (MSCs) per volume [4,5,6,7]. The MSCs of fat tissue, adipose tissue-derived stem cells (ADSCs), have the potential to differentiate into various cell lineages including the adipogenic, chondrogenic, osteogenic, neural, myogenic, and endothelial lineages [1,8,9,10,11,12,13]. Additionally, ADSCs have immunomodulatory properties and secrete growth factors crucial for cell proliferation and angiogenesis [7,14,15,16,17,18,19,20,21]. Therefore, autologous fat grafting, or lipofilling, has become one of the most frequent procedures in plastic surgery for aesthetic and reconstructive purposes [22,23,24,25]. It has been optimized since the first reports about the transfer of autologous fat in the 19th century [26,27], the first autologous fat injection with a needle in the early 20th century [28,29], and the establishment of modern liposuction in the late 20th century [30,31,32,33,34,35]. Today, lipofilling with autologous fat is a safe and relatively simple procedure, with no risk of immunogenicity and low costs [36,37,38,39,40]. However, graft survival and take rate remain hard to predict, with volume losses of up to 70% [41,42,43,44]. This poses the main challenge for the clinical application of lipofilling and autologous fat transfer in regenerative medicine. Therefore, a lot of effort has been put into developing methods to improve fat graft survival. These efforts range from the optimization of steps inherent in lipografting, such as harvesting-site choice and liposuction technique, to supplementary steps such as tissue processing or stem cell addition. Here, we will provide a short summary of factors affecting fat graft survival and present a new method of lipoaspirate processing resulting in improved tissue preparation with excellent properties for graft retention and long-term survival.

## 2. Tissue Harvesting

The choice of the harvesting site seems to have only a minor impact on graft volume retention and cell viability [35]. Among the most widely utilized donor sites, such as the abdomen, buttocks, and posterior thigh, no significant differences were found in several studies [35,45,46,47,48]. The liposuction technique, on the other hand, can greatly affect the viability and survival of grafted cells. Several factors influencing the graft take rate can be identified. Fundamentally, the dry technique, which is normally performed under general anesthesia, can be distinguished from various wet techniques where an injectant is infiltrated into the donor site prior to liposuction [49,50]. Nowadays, the most-used modification of the wet technique is the tumescent technique [32]. Here, the infiltrate contains epinephrine among the local anesthetics to minimize blood loss via vasoconstriction. It has been shown in several studies that lipografts harvested using wet techniques show improved cell viability compared to tissues obtained using a dry technique [35,51]. However, in a histological study by Agostini et al., no significant differences between samples from dry and wet harvesting techniques were found [52]. Whereas the vast majority of studies indicate that the tumescent technique is better for graft survival than other wet techniques or the dry technique, there is no agreement on the effect of different local anesthetics on the viability and differentiation potential of harvested cells. For lidocaine and other local anesthetics frequently used in tumescent techniques, a negative impact on cell viability was shown [53,54,55]. In other studies, no negative effect on graft survival was reported for the same substances [56,57,58]. These differences might be explained by different substance concentrations and durations of exposure as well as timepoints and methods for viability measurement. Another parameter affecting cell viability is the size of the harvesting cannula. Since its introduction by Coleman [33,59,60], various canula sizes and their effects on fat grafting have been evaluated. Whereas larger cannulas tend to reduce mechanical stress for harvested cells, smaller cannulas reduce insertion injuries [35,61,62,63]. It has been proposed that 17-gauge cannulas are a good compromise in terms of the viability of grafted cells and the preservation of anatomic structures [59,64]. Another parameter exerting cellular stress, and therefore influencing graft viability, is the negative suction pressure. The utilization of machines for both automated negative pressure and manual syringe aspiration have advantages and disadvantages. Especially when larger volumes of lipoaspirate are needed, an automated system is much faster, whereas everything needed for manual syringe aspiration can easily be set up if a smaller volume is sufficient. Whereas older studies especially implicate a negative effect of automated vacuum aspiration [65,66], there are some more recent studies that reported no significant differences between vacuum and syringe aspiration [67,68]. However, it is generally accepted that constant and moderate negative pressure is beneficial to reducing harvesting stress. Although many efforts have been made to optimize liposuction parameters in order to improve cell viability and graft take rate, it is important to note that several studies indicate that every kind of liposuction impairs fat graft cell viability in comparison to excised fat [65,69,70,71,72]. However, depending on the evaluation parameter, conflicting results without significant differences can be found, too [72,73,74].

## 3. Tissue Processing

Supplementary steps, such as tissue processing between harvesting and reapplication, primarily serve the purpose of the removal of non-beneficial components. These components include, among others, lipids from disrupted mature adipocytes, tumescence solution, inflammation factors, cellular debris, and regeneratively useless erythrocytes [23,35,75]. It has been shown that the removal of these contaminants is beneficial for fat graft survival [23,75,76,77]. The simplest method to achieve separation is sedimentation. Here, the lipoaspirate is allowed to separate into phases under gravitational force. An aqueous phase, being the densest phase, settles at the bottom of a given container. A lipid phase, being the least dense phase, settles as the top layer. In between is the desired adipose tissue. Sedimentation alone is sufficient to increase the percentage of target cells via separation from unwanted contaminants and to support graft survival [78]. This phase separation can be accelerated and strengthened by applying centrifugal forces. Centrifugation led to a higher concentration of adipocytes and ADSCs compared to sedimentation but did not further increase graft viability [35]. This might be related to the cell-survival hypothesis for spherical lipografts, which is discussed in more detail below [39,72,79]. Briefly, the enrichment of cells beyond sedimentation might be useless without decreasing the size of the transplanted particles. An alternative to sedimentation and centrifugation is filtration. Both automated and manual techniques are applied commonly. Although filtration techniques resulted in both the satisfactory removal of contaminants and a higher concentration in cellular components, the graft take rate was not improved in most studies [80,81,82,83,84,85].

Although sedimentation, centrifugation, and filtration remove lipids, water, and other contaminants, providing a tissue with an increased number of viable cells in general, a different approach of tissue processing is needed for enrichment with stem cells particularly. Most of the effects that are beneficial for fat graft survival depend on secreted factors [86]. Neovascularization and cellular regenerative potential have been identified as major requirements for the graft take rate [17]. Because ADSCs, like all MSCs, secrete growth factors for neoangiogenesis and cell survival, a higher number of stem cells per lipograft volume is a promising approach for take rate improvement. The obvious choice to achieve a higher stem cell number is simply the addition of culture-expanded ADSCs, a procedure known as cell-assisted lipotransfer (CAL) [87,88,89,90]. This method was shown to be effective in supporting fat graft retention [91,92,93] but includes the isolation and cultivation of stem cells, and therefore, bears the risk of regulatory restrictions. The enzymatic digestion of tissue [94] is considered a substantial manipulation, rendering the obtained cells an advanced-therapy medicinal product (ATMP) in Europe (1394/2007) and the USA (21 CFR 1271.10) [95]. The utilization of xenogeneic digestion enzymes or cultivation with xenogeneic culture sera may further prevent clinical usage of these cells [96,97]. Enzyme-free isolation techniques, such as explant cultures and defined serum-free cell culture media, could evade the problems that come along with xenogeneic substances [94]. However, the isolation and in vitro cultivation of cells might still be considered an inadmissible substantial manipulation. Additionally, the utilization of isolated and cultured autologous cells would necessitate a second surgical intervention. However, washing, rinsing, centrifuging, and filtration aren’t considered substantial manipulation. Tissue preparation obtained in this way is suitable for structural support, and is therefore considered minimally processed [39].

Mechanical processing of lipoaspirate constitutes an alternative for the further enrichment of stem cells [77]. Here, the tissue is exposed to mechanical shear stress or other physical forces, which is not substantial manipulation [91]. An example of mechanical shear stress applied to lipoaspirate is intersyringe processing [98]. For this, the lipoaspirate is shifted quickly between two syringes through a connector with a given, usually small, diameter. This leads to decreased particle size and the rupture of many mature adipocytes without affecting the small and robust ADSCs. Table 1 summarizes the different tissue-processing methods. The method of intersyringe processing using the specific device, technology, and protocol presented, and its implications, will be discussed in more detail in the next section. However, an alternative for particle-size reduction is the utilization of a strainer of defined pore size [99].

## 4. Cell-Enriched Lipotransfer

Cell-Enriched Lipotransfer (CELT) is a sequence of simple consecutive steps that result in a stem cell-enriched tissue. This processing method can be combined with any harvesting technique and has a broad range of applications. However, we will give a brief overview of the harvesting procedure commonly used by us. A 0.9% (*w*/*v*) solution of NaCl (a volume equal to the volume of fat tissue to be harvested) containing epinephrine (1:200,000) is infiltrated over approximately 15 min according to the S2K guidelines [24] using a 2.5 mm injection cannula. The lipoaspirate is harvested using a 3.8 mm cannula. We utilize a machine for automated negative pressure (Body-Jet^®^, Human Med AG, Schwerin, Germany), which allows water-jet-assisted liposuction with a constant negative pressure of less than 0.5 mbar. After the lipoaspirate is allowed to sediment, it is centrifuged at 1600× *g* for 2 min with brakes enabled. Subsequently, the tumescent layer and the lipid layer are discarded. For intersyringe processing, a connector with a diameter of 1.2 mm is used and the lipoaspirate is forced manually, 10 times in total, quickly through the connector. Afterwards, the processed tissue is once more centrifuged at 1600× *g* for 2 min and separated from the aqueous phase and the lipid phase. The total processing time is under ten minutes.

With the first centrifugation, the infiltrated tumescence solution is primarily separated from the tissue preparation. Only a small lipid layer from the adipocytes disrupted during the liposuction resulting from this centrifugation. However, it is crucial to remove this large amount of aqueous phase prior to the intersyringe processing. With the second centrifugation, the released lipids from disrupted mature adipocytes are primarily separated from the tissue preparation. Only a small aqueous layer from the adipocytes’ cytosol emerges from the second centrifugation (Figure 1). The resulting tissue contains highly concentrated ADSCs and other cells from the stromal vascular fraction in a massively reduced volume, with a reduced number of mature adipocytes and a reduced size of fat lobules. Although no individual step of the procedure is a novelty in its own right, the combination in the specified order and with the specified parameters results in a graft optimized for graft retention, graft survival, and tissue regeneration. This tissue preparation can be used directly for the treatment of low-volume defects for which it is especially suitable because it can be applied through very fine cannulas with diameters of less than one millimeter. However, it can also be used as an addition to one-time centrifuged lipoaspirate if larger volumes are needed, albeit with lower concentration of ADSCs.

We were able to demonstrate that our mechanical lipograft processing causes no substantial manipulation of the cells. ADSC viability and cytokine secretion were not compromised by the method [100]. The tissue structure is preserved, with significantly smaller particles. Additionally, the differentiation potential of ADSCs in the processed tissue was unchanged. Histologically, we were able to demonstrate that the mature adipocytes from large adipocyte aggregations were especially disrupted by the shear-force processing. Mature adipocytes adjacent to the vascular system or large extracellular matrix depositions were unaffected by the mechanical stress [101]. This is in accordance with a recently published hypothesis regarding the lobule organization of adipose tissue and its impact on regeneration potential [102]. Clinically, we were able to demonstrate excellent results with CELT lipografts for facial rejuvenation and other applications [103,104].

It is believed that transplant survival rates depend on the viability of grafted cells, neoangiogenesis, and the overall regeneration potential of the grafted cells [17,87,105,106,107,108,109]. Therefore, the higher concentration of stem cells in CELT tissue could explain the increased graft retention we observed in the clinic. However, many studies with centrifuged, and therefore, cell-enriched lipoaspirate failed to show an increased take rate in comparison to sedimented lipoaspirate [110]. Recently, a cell-survival hypothesis for spherical lipografts was proposed [39,72,79]. Therein, it was hypothesized that only a thin outer layer (100–300 µm) of the fat lobule in close proximity to oxygen and a nutrient supply can survive. In the middle zone, ADSCs may survive and proliferate but the central zone is mostly necrotic. Therefore, further stem cell enrichment by centrifugation beyond the sedimentation-induced cell concentration might be useless as long as the particle size is too big to allow for diffusion to the central zone. The ADSCs are also much more resistant to external conditions such as oxygen deprivation and radiotherapy, as shown in our studies [111,112,113]. In this cell-survival hypothesis of spherical grafting, the maximum size for fat particles is 1.5 mm before the innermost part is too far from the surface to escape necrosis [39]. However, several studies indicate that, although the particle size should not exceed a threshold that prevents diffusional supply, those remaining within the tissue structure have numerous benefits for the cells’ regenerative properties. In contrast to isolated single cells, the cells in microfragmented fat tissue show extended survival and long-lasting anti-inflammatory activity [114]. Additionally, the cell–matrix contacts and the cell–cell contacts with other cells of the stromal vascular fraction promote the proliferative and angiogenic effects of ADSCs [95,115,116]. Additionally, the physical properties of adipose tissue contribute to the therapy of inflammatory-joint-like conditions such as osteoarthritis [117].

It has been shown that cells from microfragmented fat show increased activity of cytokine and growth-factor secretion in comparison to enzymatically obtained cells [118]. This further contributes to the increased therapeutic potential of mechanically processed adipose tissue [95]. Interestingly, besides the increased secretory activity, cells obtained from microfragmentation show a higher pericyte content than the stromal vascular fraction obtained from enzymatic digestion [118]. Pericytes exist ubiquitously in vascularized tissues throughout the human body [119,120,121]. Additionally, it has long been proposed that ADSCs have a pericyte origin [122,123,124,125,126]. With a high pericyte content in mechanically processed adipose tissue, new developments in the discussion about its homologous use (“fat in fat”) and its regulatory indications might emerge. Although the debate about the universal perivascular origin of MSCs remains unresolved with opposing positions [127], the higher pericyte content might be able to expand the area of application for autologous fat grafting in homologous use.

## 5. Conclusions

In past decades, many efforts have been made to improve the take rate and survival of lipografts. The CELT method combines contaminant removal, stem cell concentration, and particle-size reduction to improve fat graft retention in a fast, reliable, and cheap way and is in accordance with all regulatory requirements. The method is evaluated regarding both the cellular characterization and the clinical outcome.

## Figures and Tables

**Figure 1 cells-11-03159-f001:**
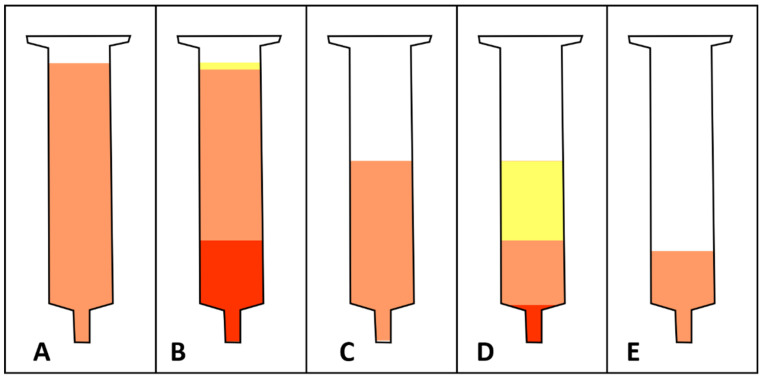
Schematic illustration of macroscopic lipoaspirate morphology after each of the four major steps for tissue processing with the CELT protocol. (**A**) After it is allowed to sediment under gravitational forces, a syringe is filled with lipoaspirate (illustrated in orange). (**B**) After the first centrifugation, a large aqueous phase settles at the bottom of the syringe (illustrated in red) and a small oily phase separates at the top of the syringe (illustrated in yellow). The size of the aqueous phase varies depending on the time the tissue is allowed to sediment. The size of the oily phase varies depending on the shear stress the harvesting method has exerted. (**C**) After the aqueous and oily phases have been discarded, shear-force mechanical processing via intersyringe processing can occur. (**D**) After mechanical processing and a second centrifugation, a small aqueous and a large oily phases separate from the tissue at the syringe’s top and bottom, respectively. (**E**) After the aqueous and oily phases have been discarded, a stem cell-enriched lipograft tissue is ready for clinical application.

**Table 1 cells-11-03159-t001:** Different tissue-processing methods and the impact on different parameters. Centrifugation and filtration are better for the removal of contaminants than sedimentation, but do not increase graft viability without particle-size reduction. Mechanical processing in combination with centrifugation combines contaminant removal with the best stem cell enrichment and graft viability enhancement without regulatory restrictions (-: no impact; *: impact; **: stronger impact; ***: strongest impact).

	Contaminant Removal	Stem Cell Enrichment	Increase in Graft Viability	Possibility of Regulatory Restrictions
Sedimentation	*	*	*	-
Centrifugation	**	**	*	-
Filtration	**	**	*	-
Cell-enriched Lipotransfer	-	***	**	***
Mechanical shear stress	-	***	**	-

## Data Availability

Not applicable.

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
