# Peer review of "Cell-Enriched Lipotransfer (CELT) Improves Tissue Regeneration and Rejuvenation without Substantial Manipulation of the Adipose Tissue Graft"

_cells, 2022, doi:10.3390/cells11193159_

Round 1

Reviewer 1 Report

The manuscript reviewed the status of the approaches used for adipose tissue harvest and processing, and cell preparation. Specifically, the authors compared the advantages and disadvantages of different approaches and discussed a new approach of cell-enriched lipotransfer and its promising in promoting tissue regeneration within adipose tissue graft.  The content in the manuscript is very informative and important in tissue regeneration fields. 

Comments: 

  • Page 3 (120-121 lines): replace “adipose tissue derived stem cells” with “ADSCs”. 
  • Page 4-Figure 1: please describe the meaning of each color in the syringe figure in the Figure 1 legend or in the Figure 1B. 
  • Inclusion of one table that lists and compares different tissue process approaches (advantages vs. disadvantages: e.g., time used for cell process, tissue viability, the regeneration potential) will increase well-structured information.  
  • The manuscript title does not sufficiently meet the content included in the manuscript. I suggest adding the detailed description in the paragraph (lines 190-199) regarding the experimental and clinical evidence showing the improvement of tissue regeneration within adipose tissue grafts. 

Author Response

The manuscript reviewed the status of the approaches used for adipose tissue harvest and processing, and cell preparation. Specifically, the authors compared the advantages and disadvantages of different approaches and discussed a new approach of cell-enriched lipotransfer and its promising in promoting tissue regeneration within adipose tissue graft.  The content in the manuscript is very informative and important in tissue regeneration fields.

We thank the reviewer for the encouraging remarks.

Comments:

  • Page 3 (120-121 lines): replace “adipose tissue derived stem cells” with “ADSCs”.
  • We changed this according to the reviewers’ suggestion.
  • Page 4-Figure 1: please describe the meaning of each color in the syringe figure in the Figure 1 legend or in the Figure 1B. 
  • We thank the reviewer for this suggestion. We described the meaning of the colors in the figure legend and this clearly improves the figure.
  • Inclusion of one table that lists and compares different tissue process approaches (advantages vs. disadvantages: e.g., time used for cell process, tissue viability, the regeneration potential) will increase well-structured information.
  • We have added a summarization table as suggested by the reviewer. However, we used not the exact parameters suggested by the author because these parameters are not applicable for all processing steps.
  • The manuscript title does not sufficiently meet the content included in the manuscript. I suggest adding the detailed description in the paragraph (lines 190-199) regarding the experimental and clinical evidence showing the improvement of tissue regeneration within adipose tissue grafts.
  • We have extended the manuscript title to reflect the experimental and clinical evidence.

We want to thank the reviewer for the supportive comments that helped to improve our manuscript.

Reviewer 2 Report

The manuscript describes the advantage of CELT, specifically with reference to a previous publications by the authors.

I feel that  the writing is superficial for a review article and lacks novel information/state of the art/comparisons/conclusions etc.. Hence all the sections of the manuscript need to be fleshed out to make this a worthwhile read. I do not think that it can be published in its current form it.

Author Response

The manuscript describes the advantage of CELT, specifically with reference to a previous publications by the authors.

I feel that  the writing is superficial for a review article and lacks novel information/state of the art/comparisons/conclusions etc.. Hence all the sections of the manuscript need to be fleshed out to make this a worthwhile read. I do not think that it can be published in its current form it.

Having a “classical” review paper in mind, the reviewer is correct, of course. However, this manuscript is supposed to be some sort of hybrid paper. So far, we have already published experimental and clinical data for our method/protocol without describing the method in detail. Here, we wanted to present the method itself without focusing on experimental data but putting the details of the method into context (the “review” part). To keep the focus on the presentation of the method, the context is not as comprehensive as could be expected from a pure review paper. The reviewer is definitely correct about this. We can discuss with the editorial office whether there is a more fitting category for this manuscript. However, rewriting the whole manuscript to meet the expectations for a pure review paper is not possible in the time given for the revision, would go beyond the size scope when still focusing on the presentation of our method, and was never the intended structure for this manuscript.

Reviewer 3 Report

As a review paper it seems that the write up of the subject should be longer and more involved. 

Authors should also note the extensive work by Choudhery et al in this area.

Please have the authors emphasize the novelty of the submitted work.

The authors should also discuss other methodology such as LipoGems that are found in the literature and in common practice.

Authors should also note whether they believe that ADSC are one and the same as MSC.

Author Response

As a review paper it seems that the write up of the subject should be longer and more involved.

Having a “classical” review paper in mind, the reviewer is correct, of course. However, this manuscript is supposed to be some sort of hybrid paper. So far, we have already published experimental and clinical data for our method/protocol without describing the method in detail. Here, we wanted to present the method itself without focusing on experimental data but putting the details of the method into context. To keep the focus on the presentation of the method, the context is not as comprehensive as could be expected from a pure review paper. The reviewer is definitely correct about this. We can discuss with the editorial office whether there is a more fitting category for this manuscript. However, rewriting the whole manuscript to meet the expectations for a pure review paper is not possible in the time given for the revision, would go beyond the size scope when still focusing on the presentation of our method, and was never the intended structure for this manuscript.

Authors should also note the extensive work by Choudhery et al in this area.

We have added works from Choudhery et al. regarding the filtering of lipoaspirate (New reference #86) and regarding the pre-enrichment with ex-vivo expanded ASCs (New references #91-93).

Please have the authors emphasize the novelty of the submitted work.

We have added a statement that summarizes the novelty of the presented method (a few lines above figure 1)

The authors should also discuss other methodology such as LipoGems that are found in the literature and in common practice.

Our method is a combination of purification and homogenization. LipoGems takes a similar approach. However, this procedure is offensively advertised for not needing strong mechanical forces but utilizing some sort of strainer for tissue homogenization, whereas the strong mechanical forces are an integral part of our method. We have added the kind of processing utilized by LipoGems to the manuscript. However, we have no interest in judging a commercial product directly.

Authors should also note whether they believe that ADSC are one and the same as MSC.

In our opinion, ADSC are MSC. However, there are other MSCs than ADSCs (for example dental pulp derived mesenchymal stem cells). We have carefully revised our manuscript for occasions where such a notion could be needed and we hope that we could clarify our perspective on the matter.

Round 2

Reviewer 2 Report

I appreciate the author's clarification about the intent of the manuscript. I believe that that the manuscript in its current form does not qualify to be published as a review article. It probably can be considered as a hybrid methods/review article, but I would leave that to the Editor's judgement.